# Curious Dichotomies of Apolipoprotein E Function in Alzheimer’s Disease and Cancer—One Explanatory Mechanism of Inverse Disease Associations?

**DOI:** 10.3390/genes16030331

**Published:** 2025-03-12

**Authors:** Claire M. Perks, Rachel M. Barker, Mai Alhadrami, Omar Alkahtani, Emily Gill, Mary Grishaw, Abigail J. Harland, Peter Henley, Haonan Li, Ellie O’Sullivan, Gideon Stone, Xiaoyu Su, Patrick G. Kehoe

**Affiliations:** 1Cancer Endocrinology Group, Bristol Medical School, Learning & Research Building, Level 2, Southmead Hospital, Bristol BS10 5NB, UK; mdrmh@bristol.ac.uk (R.M.B.); oc21138@bristol.ac.uk (M.A.); fp23869@bristol.ac.uk (O.A.); vw19109@bristol.ac.uk (E.G.); ah15974@bristol.ac.uk (A.J.H.); av20901@bristol.ac.uk (H.L.); xiaoyu.su@bristol.ac.uk (X.S.); 2Cerebrovascular and Dementia Research Group, Bristol Medical School, Learning & Research Building, Level 2, Southmead Hospital, Bristol BS10 5NB, UK; mollie.grishaw@bristol.ac.uk (M.G.); peter.henley@bristol.ac.uk (P.H.); eo17170@bristol.ac.uk (E.O.); gideon.stone@bristol.ac.uk (G.S.)

**Keywords:** apolipoprotein E, Alzheimer’s disease, cancer, inverse association

## Abstract

An apparent “inverse” relationship exists between two seemingly unconnected conditions, Alzheimer’s disease (AD) and cancer, despite sharing similar risk factors, like increased age and obesity. AD is associated with amyloid beta (Aβ) plaques and neurofibrillary tau tangles that cause neural degeneration; cancer, in contrast, is characterized by enhanced cell survival and proliferation. Apolipoprotein E (ApoE) is the main lipoprotein found in the central nervous system and via its high affinity with lipoprotein receptors plays a critical role in cholesterol transport and uptake. ApoE has 3 protein isoforms, ApoE E2, ApoE E3, and ApoE E4, respectively encoded for by 3 allelic variants of *APOE* (ε2, ε3, and ε4). This review examines the characteristics and function of ApoE described in both AD and cancer to assimilate evidence for its potential contribution to mechanisms that may underly the reported inverse association between the two conditions. Of the genetic risk factors relevant to most cases of AD, the most well-known with the strongest contribution to risk is *APOE*, specifically the ε4 variant, whereas for cancer risk, *APOE* has not featured as a significant genetic contributor to risk. However, at the protein level in both conditions, ApoE contributes to disease pathology via affecting lipid physiology and transport. In AD, Aβ-dependent and -independent interactions have been suggested, whereas in cancer, ApoE plays a role in immunoregulation. Understanding the mechanism of action of ApoE in these diametrically opposed diseases may enable differential targeting of therapeutics to provide a beneficial outcome for both.

## 1. Introduction

### The Molecular Biology and Function of ApoE

In humans, Apolipoprotein E (ApoE) is a 299-amino acid long protein (34 kDa) that is a member of a group of amphiphilic exchangeable apolipoproteins. It is produced in cells of the liver, central nervous system (CNS), adipose tissue, and kidneys as well as monocytes and macrophages [1]. ApoE is vital to numerous physiological functions, especially lipid metabolism and cholesterol homeostasis in the CNS [2,3]. Unlike other apolipoproteins, in aqueous solutions, ApoE is composed of two distinct and independently folded domains: an amino-terminal-based receptor binding region and a carboxyl-terminal-based lipid binding region [4]. The amino-terminal domain contains four helices stabilized by both a tightly packed hydrophobic core with leucine interactions and salt bridge interactions at the mostly charged surface. Several basic residues clustering on the surface of the 4th helix are important for receptor binding [5]. The lipid binding region containing the carboxyl-terminal domain has a role in stabilizing the lipid-free protein in a tetrameric form [6].

ApoE is encoded for by the gene *APOE*, which has been mapped to chromosome 19q13.32, alongside *ApoC*, that encodes for another exchangeable lipoprotein. *APOE* variants arise from two relatively common point mutations, involving cysteine–arginine interchanges at residues 112 (rs7412 C/T) and 158 (rs429358 C/T). The resultant mutation permutations give rise to three *APOE* alleles (*APOE* ε2, ε3, and ε4) that are each and respectively responsible for the production of protein isoforms: ApoE E2, ApoE E3, and ApoE E4 (Figure 1) [1,7]. ApoE E2 has cysteine residues at amino acid sites 112 and 158; ApoE E3 has a cysteine and arginine residue at sites 112 and 158, respectively, while ApoE E4 has two arginine residues at the two sites [7]. These mutations account for some distinct structural differences between the three ApoE isoforms, that can affect their physiological function, interactions with receptors, and roles in various diseases [8].

ApoE acts as a ligand for a group of receptors referred to as the low-density lipoprotein (LDL) receptors. There are seven known receptors in this family, all of which share structural properties that allow it to bind ApoE [9]. The role of each receptor, when bound to ApoE in the CNS, where there has been extensive research due to the involvement of ApoE in AD (see below), as well as any variability of function due to isoform-specific interactions, is summarized in Table 1.

## 2. A Brief Introduction to Alzheimer’s Disease

Alzheimer’s disease is perhaps one of the most known conditions that has strong ApoE involvement. It is a progressive neurodegenerative disorder, described as a condition of memory loss that is caused by marked loss of neurons and resultant brain atrophy. Alzheimer’s disease is by far the largest cause of dementia [24], which is an umbrella term for a number of related conditions, caused by different or combinations of neurodegenerative processes, that involves the progressive loss of cognitive function and eventually the ability to function independently and requires extensive levels of full time care [24,25,26]. Dementia collectively, but especially AD due to its individual prevalence, poses very significant global health, social, and economic challenges. Within the UK, the anticipated health care cost for 2024 is likely to be approximately £42 billion. This relates to nearly 1 million individuals affected in 2024 while estimates are that the number of cases is expected to rise to 1.4 million by 2040 [27]. In the US, almost 7 million individuals are thought to be affected, and prevalence is projected to rise to 13.8 million by 2060. The total cost of dementia care, including healthcare, long-term care, and hospice services, are estimated at 360 billion USD in 2024 [28].

There are also important differences between sex and ethnicity regarding AD, where men have a lower risk of developing AD (1 in 10) compared to women (1 in 5). This stark difference was initially attributed to women’s longer life expectancy; however, emerging evidence suggests that biological factors, such as hormonal differences (e.g., estrogen levels), may play a larger role than might have been previously appreciated [29]. Furthermore, black African, Caribbean, and south Asian populations face a higher risk of AD compared to white populations, largely due to a higher prevalence of various modifiable risk factors, such as hypertension, dyslipidemia, obesity, and diabetes, which likely contribute to faster rates of progression in some groups [30]. However, more accurate studies of progression, usually based on biomarkers are limited and problematic as ethnic minority groups remain heavily underrepresented in these studies and clinical trials [31].

## 3. A Brief Overview of Cancers of Older Age

The relationship between increased aging and cancer development is thought to relate to longer exposure times to relevant risk factors, in combination with age-related reduced efficiencies in cell repair mechanisms, where cellular damage and tumor formation may be more likely to occur. Additionally, aging leads to a decline in mitochondrial function, which results in the accumulation of ROS. Furthermore, telomere shortening with age causes cellular senescence, leading to chronic inflammation and further linking the association between aging and cancer [32]. The diagnosis of cancers is often prompted by the development of symptoms that are investigated by physical examination, laboratory tests on various sample types (e.g., biopsy), and imaging techniques to try and achieve the greatest certainty as to an accurate diagnosis, severity and level of progression of disease [32]. Tumors can be either benign or malignant; both are a result of abnormal cell division [33]. Benign tumors remain restricted to their original site and cannot invade surrounding tissues [33]. Malignant tumors, on the other hand, can invade surrounding tissues (Figure 2) and are capable of spreading to other parts of the body through the circulatory or lymphatic systems [34].

The high proliferative nature of cells in cancer contrasts markedly (Figure 3) with Alzheimer’s disease whereby extensive neuronal cell death and brain atrophy are key characteristics [26]. However, both of these conditions share age as a major risk factor and have similar risk factors, including obesity and co-morbid metabolic disorders, such as type 2 diabetes.

## 4. Curious Inverse Associations Between Alzheimer’s Disease and Cancer

With increasing aging and life expectancy in global populations [35], cases of AD [36] and cancer are rising [37]. However, perhaps counter-intuitively, various studies investigating the corresponding relationships between these diseases report that their incidences are inversely correlated. In other words, levels of cancer appear to be significantly lower than expected in people who received a diagnosis of Alzheimer’s disease [38,39,40,41,42,43,44] with one estimate putting this reduction as much as 70% [45]. Conversely rates of AD are significantly lower, by just over half (55%) in one study [46], in people whom have received a diagnosis of cancer [38,42,43,47,48,49].

Inverse associations have been reported across a variety of populations and cancer types, sometimes not consistently, but have been supported by several meta-analyses [50,51,52,53]. Due to the high mortality rate of both diseases, one regularly made and understandable explanation for these curious inverse associations is that it is a manifestation of ‘survival bias’, which might reasonably explain reduced rates of one disease following the other disease. However, various additional modelling that has been regularly applied in these studies has failed to demonstrate survival bias as a major contributor and the association was not diminished by excluding data from people who died before the age of 80, indicating survival bias does not fully explain this association [38]. Furthermore, the inverse associations are equally present in studies on cancers with a lower mortality risk, such as nonmelanoma skin cancer, where the potential influence of mortality bias is less [52,53].

One meta-analysis also sought to explore the potential of ‘diagnosis bias’ as an explanation for this apparent phenomenon but similarly found that the inverse associations cannot be explained by this, as the studies determined to be most susceptible to diagnosis bias, i.e., those without regular memory assessment or health checks, which could lead to underdiagnosis of both conditions, made the inverse association weaker when included in the analysis [54]. Similarly, postmortem studies of people who had died with a diagnosis of AD showed less previous cancer incidence [54].

There is also some evidence that suggests these two diseases may have some genetic correlations [55]. Mendelian Randomization (MR) analysis has shown that risk of AD is significantly inversely associated with single nucleotide polymorphisms (SNPs) most strongly associated with cancer types shown previously to have an association with AD through epidemiological evidence [56]. This indicates there is some significant interaction occurring at the genetic level and evidence for a vertical pleiotropic effect [57], while another MR study suggested associations between AD and cancer that might be mediated through the ApoE receptor VLDLR [58]. Transcriptomic meta-analyses of microarray gene expression data have also shown there to be a significant overlap in oppositely regulated, differentially expressed genes in lung, prostate, and colorectal cancer and AD [59]. This suggests there could be common molecular pathways involved in both diseases but with dysregulation in opposing directions, causing the differing pathological progression and perhaps influencing the different effects on cells between the two [60].

As an example, if dysregulations occurred that had a bearing on the cell cycle, it may push cell activity towards a propensity for apoptosis, like that seen in AD or, alternatively, increased cell proliferation, leading to tumorigenesis in cancer. Analyses along these lines have specifically highlighted diverging regulation of Wnt signaling, Pin1 and p53 [59]. Pin1 is a cell cycle enzyme [61], which is overexpressed in a variety of tumor cells [44], but a reduction in soluble Pin1 in AD hippocampal brain tissue has been found [62]. Pin1-knockout mice have shown resistance to breast cancer induced by Neu and Ras [63] but found to develop AD-like pathology, such as neuronal degeneration, tau hyperphosphorylation, and filament formation [64].

p53 is involved in cell cycle arrest and is encoded by the *TP53* gene, which is the most frequently mutated gene in cancer [65]. Mutated or inactivated p53 has been connected to several cancer types [66], whereas high p53 expression has been reported in AD postmortem brain tissue [67], particularly surrounding Aβ plaques [68]. Conversely, hyperactivation of canonical Wnt signaling, which is involved in tissue maturation and homeostasis, has been linked to cancer [69,70], whereas evidence suggests this has protective effects against Aβ toxicity in hippocampal neurons [71].

These three molecular pathways are intimately connected in terms of their normal functions. Pin1 regulates a variety of tumor suppressor proteins, including p53, by effecting its ability to respond to DNA damage [72]. p53 interacts with the Wnt signaling pathway [73]. These have been proposed as potential molecular mechanisms’ to explain the apparent inverse correlations between AD and cancer [74,75,76,77,78] but little has been investigated surrounding *APOE*, despite the fact that its product (ApoE) modulates these molecules and has such a profound effect in AD.

ApoE E4 directly inhibits canonical Wnt signaling [79] and mice expressing human *APOE* ε4 have decreased Pin1 mRNA levels in the entorhinal and parietal cortex but increased levels in the hippocampus compared to *APOE* ε3 mice [80]. People with melanoma who are carriers of the *APOE* ε4 allele have higher survival rates than those with *APOE* ε2 [81]. Similarly, mice bearing the human *APOE* ε4 variant show enhanced anti-tumor immune activation and reduced melanoma progression compared to those with *APOE* ε2 [81]. Despite these interesting findings there remains a significant gap in knowledge as to what might be the mechanisms behind how variation in *APOE* can mediate these effects and thus be of relevance to the increasing number of reported inverse associations between AD and cancer. There may be some clues based on findings that both LDLR and LRP1 are expressed in skin cells and important in skin homeostasis [82,83]. Amyloid precursor protein (APP) expression also appears relevant in some skin aging conditions [84] that are altered in familial forms of AD [85], suggesting again how the co-location of APP, ApoE, and ApoE receptors may be important. Indeed, perturbation of APP biology and processing, especially large reductions in γ-secretase, that is partly responsible for the formation of Aβ peptide, may have consequences for tumorigenesis [86] and where Aβ may, by virtue of its apoptosis-promoting effects, be an unlikely ally against some types of tumors [87].

## 5. ApoE in Alzheimer’s Disease

The aforementioned association between *APOE* variation and AD was first hinted at in early genetic linkage studies in families with some, although not consistent, evidence of Mendelian inheritance that had an onset of 60 years [88], with follow-up work refining this signal to be coming from or near the *APOE* locus [89]. The proposition of *APOE* as a likely candidate risk locus for AD was further supported by findings of reported interactions in vitro between ApoE and Aβ, but also evidence that people with *APOE* ε4 alleles, homozygotes in particular, appeared to have more pronounced vascular and Aβ plaque deposits than *APOE* ε3 carriers [90]. Over time there has been a vast number of supportive studies, and it is now widely accepted that the *APOE* ε4 variant is the largest genetic risk factor for sporadic LOAD and which has been borne out in recent GWAS [91,92,93].

Notably, whilst heritability estimates for *APOE* variants for AD have been determined in many different studies, they are very variable due to the direct and indirect influence of many different factors [94,95]. The physiology of lipid processing and transport within the cerebral parenchyma differs from outside the CNS [96]. In the brain, ApoE is primarily synthesized and secreted from non-neuronal cells, such as astrocytes and microglia, and is the main protein responsible for cholesterol and other lipid transportation to neurons. There remain questions as to how the *APOE* ε4 variant/ApoE E4 isoform increases AD risk. In 2021, the *APOE* E4 haplotype was shown to have a significant influence on the abundance of a protein critical for maintaining mitochondrial function, the mitochondrial import receptor subunit TOMM40 [97]. However, the majority of reports suggest its impact could be via Aβ-dependent and Aβ-independent mechanisms [96,98] or perhaps a combination of the two. Despite a strong involvement, a lack of clarity regarding the mechanism may explain why, despite the very strong risk posed in people, treatments targeting ApoE have not been progressed to the same level as perhaps some others [99,100].

Much of the evidence arose from early human tissue studies showing that ApoE was associated with amyloid plaques and that ApoE E4 likely promoted Aβ aggregation over other ApoE isoforms [90,101,102,103], which was supported in cellular and mouse model-based experiments [96,104,105,106,107]. These findings suggested that ApoE isoforms may result in differential levels of binding to and/or mediating Aβ clearance from the brain via ApoE receptors. Some studies have suggested that ApoE E4 was less efficient in the clearance of Aβ-containing complexes due to weaker binding interactions with some of its receptors [98,108]. An alternative mechanism suggested has been that ApoE and Aβ compete for binding to the receptors but that ApoE E4 has a stronger affinity for these receptors, thereby reducing the opportunities for Aβ clearance [98,109,110].

Given that the balance of research for several decades has been heavily toward strategies that target Aβ production and clearance as the most promising way of slowing, preventing, or treating AD, it might also explain why progression of ApoE-based intervention strategies has been less prominent. Similarly, it is possible that if the role of ApoE has been interpreted mainly as having only a moderating (secondary) role to the key driver of disease (i.e., Aβ), then a strategy to focus on what is perceived to be the central disease-causing determinant makes some sense. However, there have been some and limited pursuits of ApoE intervention [87] with one gene-therapy approach using LX1001 on *APOE* ε4 homozygotes having completed a Phase I safety study (NCT03634007), and this is now in a further longer term follow-up study (NCT05400330, for further details see: clinicaltrials.gov).

According to some interpretations, the lack of ApoE interventions and/or their success to date, as well as the fact that Aβ is observed in aged individuals with no cognitive impairment, has cast some doubt on the Aβ hypothesis with some investigators [111]. However, at the time of writing, three current anti-Aβ intervention strategies have been developed (Aducanumab, Lecanemab, Donanemab) and received varying degrees of approval for use from various regulatory bodies. Aducanumab, was controversially approved by the US-based Food & Drugs Agency (FDA) but subsequently not approved by the agencies in Europe (European Medicines Agency; EMA) or the UK (Medicines and Healthcare products Regulatory Agency; MHRA), and in January 2024, Biogen (the commercial providers) announced they would no longer pursue it as a product. Lecanemab, under the name of Leqembi, produced jointly by Eisai and Biogen was approved in the US (July 2023) and subsequently Japan. In the second half of 2024, both the MHRA and the EMA approved the prescription of this but only to people with earlier (milder) stages of AD and in people who were carrying only one or no copies of *APOE* ε4. Donanemab, the newest, received FDA approval July 2024 for early AD (including mild cognitive impairment with supportive imaging evidence) and received UK MHRA approval in October 2024 with similar exclusion criteria proposed around the *APOE* carrier status.

The variable and restricted approvals by some agencies for these drugs reflects ongoing concerns around adverse events observed in trials called Amyloid-Related Imaging Abnormalities (ARIAs). These are areas of swelling (ARIA-E) and/or hemorrhage (ARIA-H) in the brain, some of which are symptomatic and some resolve over time [112,113], but there are still unclear complex interactions between ApoE and Aβ to be clarified, which may inform new opportunities to ensure more people with AD can gain access to the new treatments.

Beyond the Aβ focus, the effects of ApoE isoforms on lipid physiology and transport in the brain may also be pertinent to the pathogenesis of AD [96,98]. Compared to the most commonly occurring ApoE isoform, E3, the ApoE E4 and E2 variants have lower and higher affinities for small phospholipid rich HDLs, respectively [57,96], while ApoE E4 was less efficient at transporting cholesterol in some contexts [114]. Since these lipids are vital constituents of cell membranes in general, they are important for normal neurite growth and neuronal homeostasis. Therefore, differences in ApoE isoform-dependent changes in lipid transport could affect neuronal function and viability, and by extension, make ApoE E4 carriers more susceptible to AD [96,98,108]. This may be an over-simplification given that there is likely a spectrum of potential physiological and pathophysiological effects of ApoE isoforms on the various and different cell types throughout the CNS, which is beyond the scope of this review (for a review see [98]) but is still important to note because despite differences, there may yet be shared properties and levels of vulnerability in CNS cells, while there would be another level of complexity offered around how ApoE isoforms interact with and influence the behavior of cells in the CNS and in the periphery.

## 6. ApoE in Cancer

For the purposes of this exploratory review and to ensure it is manageable from a readers’ perspective, we have opted to focus on a small number of cancers that are most relevant to the age ranges associated with Alzheimer’s disease and are hormone sensitive to allow for some balanced exploration across the sexes.

### 6.1. Prostate Cancer

Prostate cancer (PCa) is most commonly diagnosed in men between 70 and 74 years old, with risk of development increasing above the age of 50 years. In the UK, each year, one-third of cases are diagnosed above the age of 75 years. Incidence rates in the UK have increased by 53% since the 1990s and are projected to continue rising in the coming years. Approximately 78% of men diagnosed with PCa in England survive for 10 years or more, with higher survival rates in those diagnosed before the age of 75 [115].

The etiology of prostate cancer (PCa) includes genetic and environmental risk factors. High levels of circulating cholesterol are an important factor in clinically aggressive disease forms of PCa, and the *APOE* ε4 allele has been investigated as a genetic risk factor for PCa [116,117,118]. Despite initial evidence in a small study [119] that *APOE* ε4 allele frequency appeared elevated in PCa patients and that patients with earlier disease onset appeared to be homozygous, there have been few replications. Slightly larger studies have failed to find significant differences in *APOE* ε4 allele distribution between patients (*n* = 230) and age-matched controls (*n* = 798) [120]. Similarly, the retrospective comparison of PCa patients (*n* = 1169) with non-diseased controls (*n* = 1233) found no association among *APOE* genotype, PCa progression, metastasis, and mortality [121]. Another small study proposed a contradictory finding that the ε4 allele reduced the risk of PCa, finding that *APOE* ε3 homozygotes were more prevalent in patients (*n* = 68) compared to controls (*n* = 78) [122]. While there have been inconsistent findings in smaller studies, *APOE* has not come through as a strong genetic risk factor in GWAS of PCa [123].

In contrast to the genetic risk, a more consistent observation in PCa patients is that they appear to have characteristically higher circulating total, HDL, and LDL cholesterol than controls [116,124]. It remains possible therefore, that a more complex and subtle interaction between diet and *APOE* ε4 inheritance may be relevant in PCa risk and or progression [117]. Indeed, this is further supported by evidence that statins that are used to lower cholesterol might be protective in prostate cancer [125], yet levels of ApoE were found to be quite different for *APOE* ε3 and ε2 carriers compared to *APOE* ε4 carriers who incidentally displayed more modest changes in various cholesterol targets (total cholesterol, LDL, and HDL) compared to the *APOE* ε4 non-carriers [126].

A closer look at the ApoE protein shows a stronger recognition of its potential importance because it has been repeatedly included as a key component of biomarker panels predictive for PCa prognosis, behavior, and recurrence [127,128,129,130,131,132]. In biological models of PCa, ApoE abundance correlates with tissue Gleason score, and expression is higher in aggressive cell lines (PC3) [133]. Similarly, Xia et al. (2023) showed high *APOE* expression in PC3, LNcap, and DU145 PCa cells relative to normal prostate epithelial cells [127]. The contribution of ApoE to an aggressive phenotype was further displayed by the reduced proliferative and migratory capacity of a PCa cell line upon gene silencing with *APOE* siRNA [134].

In contrast to many studies showing elevated ApoE abundance in PCa, one study showed a 24.6% reduction of overall *APOE* expression in PCa tissue compared to normal tissue from the same prostate (*n* = 69) [135]. Owing to the role of ApoE in reverse cholesterol transport, the authors concluded that lower *APOE* expression would lead to lower ApoE abundance in PCa tissue. Altered cholesterol efflux as a result was proposed to result in PCa tissue cholesterol accumulation, something previously shown to progress PCa [135].

A notable observation amongst many of these important findings was the lack of investigation into the *APOE* genotype or discrimination between isoforms in the measurements of ApoE for their potentially variable contribution to what was observed [118,136] and where further research would be very beneficial. The few studies that incorporated *APOE* genotyping in clinically aggressive models of PCa (PC3 and DU145 cells) and hormone-resistant PCa patients suggested that the *APOE* ε2/ε4 genotypes contributed to a more advanced disease [137,138] and similarly exhibited higher cellular cholesterol retention [137,139]. It may be no coincidence then that hypercholesterolemia, which increases the risk of PCa [140], is associated with the *APOE* ε2 genotype [141], where the aforementioned use of statins, often used to treat hypercholesterolemia, is protective [125,126], high cholesterol seen in people with AD [142], where *APOE* ε4 is quite common might also be an important consideration to explore further.

Secreted ApoE was recently reported to induce the senescence of infiltrating neutrophils [143], providing a potential mechanism by which ApoE could contribute to the immunosuppressive microenvironment that is seen in PCa [143]. In mouse models with gastric, colorectal, and hepatocellular carcinoma, administration of immune checkpoint inhibitors (α-PD-1 (programmed death 1) and αTIGHT (T cell immunoreceptor with immunoglobulin and ITIM domain) antibodies) displayed greater tumor reduction in *APOE −/−* than *APOE +/+ mice.* Moreover, in *APOE +/+* mice, the addition of a ApoE competitive inhibitor (COG133TFA) to the combination enhanced the antitumor effect [144]. In a separate study, implantation of mouse PCa cells into the prostate of an *APOE* knockout (KO) mouse grew more slowly than wild-type comparators [145], which coincided with fewer M2 macrophages being detected in the *APOE* KO prostate mice, reinforcing the possible importance of ApoE in the immune environment of PCa [145].

Recently, it has become apparent that ApoE may be of particular importance in castration-resistant forms of PCa. Following short-term administration of diethylstilbesterol (a synthetic form of estrogen), a subgroup of castrate-resistant PCa patients showed increased survival [146]. To delineate the mechanism of action, administration of diethylstilbesterol to adult rats resulted in reduced ApoE secretion from the liver and lower circulating cholesterol as a result [147]. Again, these results are consistent with the observation that lower circulating cholesterol is protective against aggressive PCa development [116], and thus, ApoE reduction via diethylstilbesterol could contribute to cholesterol lowering. Moreover, in PCa patients undergoing radiotherapy (*n* = 12), mean serum ApoE levels increased in patients over 21 days [148], and ApoE levels were greater in those who experienced treatment-related fatigue, which is consistent with a previously observed neurodegenerative mechanism in mice [149]. Yet again, a key absence in these studies is consideration of *APOE* variation and what might be nuanced mechanisms caused by ApoE isoforms and should be a focus for further research.

### 6.2. Breast Cancer

Breast cancer is the most diagnosed cancer in women, though it rarely affects men, accounting for less than 1% of cases [150]. Many factors contribute to its development, with aging being one of the most significant, and where most cases occur in women. In England, about 76% of breast cancer patients survive for 10 years or more, particularly when the cancer is detected at an early stage [115].

A major hallmark of breast cancer progression, like PCa, is dysregulation of lipid metabolism, whereas previously mentioned, ApoE plays a critical role [151]. In breast cancer, ApoE may be more intricately linked to cancer development [152] since inheritance of *APOE* ε2, ε3, and ε4, each appear to exert distinct effects on susceptibility and progression [152]. A meta-analysis by Saadat (2012) suggested that the *APOE* ε4 allele posed a relatively low risk for breast cancer susceptibility, particularly in Asian populations [153]. Another larger meta-analysis found a contradictory significant association between the *APOE* ε4 allele and increased breast cancer risk in Asian populations [154]. Liu and colleagues also observed that the *APOE* ε2 allele had a protective effect, reducing the risk of breast cancer [154]. These suggest that genetic and ethnic factors (perhaps including diet) may influence the relationship between *APOE* variants and breast cancer risk and incidence. One other possible consideration is the additional cognitive impairments, such as deficits in memory, processing speed, attention, and executive functions, which are commonly observed in breast cancer patients post-chemotherapy, referred to as the ‘chemobrain’ [155]. Studies have indicated that *APOE* ε4 carriers undergoing endocrine therapy tended to have worse attention and learning abilities even three to six years after treatment [156,157]. This would be consistent with associations between *APOE* ε4 carriers and cognitive deficits in AD and may relate to some neurodegenerative processes [91,92,93,158]. However, *APOE* has not featured strongly in breast cancer GWAS to date [159], so the extent of genetic risk afforded by *APOE* in breast cancer may still be weak.

Like PCa, ApoE has been suggested as a promising diagnostic and prognostic biomarker for breast cancer. The measurement of ApoE levels has been proposed as a marker for identifying breast cancer patients and monitoring disease progression. Indeed, elevated ApoE levels in patient serum have been correlated with advanced disease stages and poorer prognosis [160]. Notably, ApoE was also shown to affect tumor dynamics in its modulation of breast cancer cell proliferation and differentiation [161]. Specifically, ApoE promoted growth in hormone receptor-positive cells but inhibited proliferation in more aggressive subtypes, highlighting the there could be nuanced dose dependent or subtype-specific effects [161]. These differential effects may also relate to changes in serum triglyceride levels, which in turn reduces the levels of sex hormone-binding globulin, potentially altering hormone availability and activity [162]. Moreover, ApoE from macrophages induced apoptosis in breast tumor cells, further emphasizing a likely complex role in cancer development and progression [163].

In other experimental studies, ApoE expression enhanced proliferation in hormone-sensitive cell lines like MCF-7s while reducing migration and proliferation in triple-negative breast cancer cells [161,163]. The same study reported that MCF-7 cells expressing ApoA-I and ApoE and implanted into athymic nude mice significantly promoted tumor growth [161,163]. These findings suggest that the cellular microenvironment and hormones may be co-dependent factors in the role of ApoE in breast cancer.

### 6.3. Colorectal Cancer

Research indicates a notable gender disparity in colorectal cancer (CRC) incidence and mortality, with men being approximately 1.5 times more likely to develop and succumb to the disease compared to women [164]. While most CRC cases are diagnosed in individuals over the age of 50, emerging evidence suggests a concerning trend of increasing incidence among younger populations, potentially attributable to factors, such as obesity, lifestyle factors, and poor dietary habits [165]. In the UK, according to a leading cancer research charity (Cancer Research UK), the five-year survival rate for CRC patients is approximately 60% [115].

The common *APOE* variants have been associated with colorectal cancer (CRC) risk. In smaller studies, Kervinen et al. (1996) [165] compared 135 patients with colonic adenomas, 122 patients with colon cancer, and 199 randomly selected controls and found that the *APOE* ε4 allele was less frequent, suggesting that other *APOE* variants might be associated with higher risk in the development of proximal colonic adenomas and CRC [166]. Another study involving 206 CRC patients and 353 healthy controls demonstrated that individuals with the *APOE* ε2/ε3 genotype were at a higher risk of developing CRC compared to those with the most common ε3/ε3 genotype, while male patients with the ε2/ε3 genotype exhibited a greater prevalence of advanced CRC (Duke stages C&D) [166]; thus far, there have not been any strong signals from GWAS of CRC [167].

As with PCa and breast cancer, ApoE has been proposed as a prognostic factor in CRC [168,169]. Patients with low levels of ApoE exhibited significantly better overall and disease-free survival rates. ApoE has also been proposed as a potential biomarker for liver metastases [168] and similarly for lung metastases in CRC patients [170].

ApoE has been shown to have roles in promoting and modulating CRC progression in animal and cell studies. ApoE-deficient mice exhibited increased sensitivity to inflammatory stimuli and a heightened susceptibility to CRC in an Azoxymethane (AOM)/Dextran Sodium Sulfate (DSS)-induced CRC model [171]. Conversely, in cellular studies, overexpression of *APOE* enhanced proliferation and migration in colon cancer HCT-116 and HCT-8 cells. The Jun-ApoE-LRP1 axis was proposed as a key driver of CRC cell invasion and metastasis. Silencing of Jun and LRP1 effectively inhibited the pro-migratory effects of ApoE, further highlighting its role in CRC progression [169], but as has been seen in other cancer types, in studies of this type, *APOE* genotype or examination of individual ApoE isoforms was not investigated.

In CRC there has been more attention given to ApoE receptors. Deletion of LRP1, was associated with poorer survival outcomes in patients [172]. However, high LRP1 expression has been linked to worse prognosis following radiotherapy, suggesting its potential as a marker for distinguishing CRC from radiotherapy-resistant cells [173]. In contrast, LDLR deletion in CRC correlated with shorter survival rate [174], while overexpression of VLDLR inhibited CRC cell proliferation and migration, with its expression negatively regulated by miR-200c [175]. Since ApoE is a common ligand for all these receptors that have diverse roles in CRC contexts it is clear that further work is needed to clarify what might be ApoE isoform dependent effects but also the behavior of each receptor in relation to CRC and normal cell function in the colon.

### 6.4. Ovarian Cancer

Ovarian cancer is more commonly diagnosed in older women, with the highest incidence rates seen in those aged 75–79, with a 5-year survival rate of 45% [115]. In ovarian cancer, overproduction and higher levels of ApoE are thought to be important in more aggressive and metastatic forms of ovarian cancer, including serous carcinomas, considered the most diagnosed and aggressive form of ovarian cancer. This ApoE overexpression has been validated in cell line studies, tissue immunohistochemistry, blood serum samples, and in vivo mice studies [176,177,178,179].

Perhaps, unsurprisingly, *APOE* expression has also been shown to be increased in malignant ovarian tumors compared controls [180]. A pan-cancer study by Yu et al. that analyzed *APOE* in The Cancer Genome Atlas (TCGA) database suggested that *APOE* was most frequently amplified in ovarian, breast, and uterine cancers. Alongside this, phosphorylation of *APOE* at the S147 locus is decreased in these cancers, differing to controls in ovarian and breast cancers [181], although the mechanistic relevance remains unclear. This supports suggestions that there may be a distinct association between *APOE* expression, and thus ApoE synthesis, in malignant ovarian tissue and estrogens [182], yet, thus far, there is no strong evidence from GWAS of ovarian cancer that APOE variation may contribute to this or pose a significant or clear increase in risk [183,184,185].

The overproduction of ApoE in ovarian cancer cells may be essential for cell proliferation and the survival of the tumors [186]. ApoE expression is increased in the more aggressive tumor types [177], while knockdown of *APOE* in ovarian cancer cell lines resulted in cell cycle arrest and apoptosis. The latter observations were proposed to be due to ApoE–ApoE (LDL family) receptors that have downstream effects on survival and growth of the ovarian tumor cells [180]. Indeed, ApoE was found to be expressed on stromal fibroblasts on the borders of ovarian tumors, and these cells ordinarily interact with LRP5 receptors on the tumor cell surface, posing another mechanism that allows ApoE to induce proliferation of the cancer cells [187].

There might be, as with PCa, evidence of a role for ApoE and the immune response in ovarian cancer. Increased levels of ApoE potentially induce senescence in neutrophils, through ApoE-mediated activation of triggering receptor expressed on myeloid cells 2 (TREM2), which has recently been associated with a worse outcome in ovarian cancer patients [186] but is also of interest in AD where it is a recognized risk gene and where its role may also have an immune-related function [188,189]. In addition, the pan-cancer study by Yu et al. suggested that the elevated levels of ApoE may influence the infiltration of CD8+ T cells, with a significant association identified between the expression of ApoE and CD8+ T cells in ovarian cancer, alongside cervical, uterine, and Her2+ breast cancer [181].

## 7. Is ApoE a Mediator of Inverse Associations Between AD and Cancer?

There would appear to be some clear-cut differences in the levels of evidence in support of the role of *APOE* in relation to risk between AD and various cancers. In AD, there is irrefutable evidence of the role of *APOE* variation in the determination of risk of developing AD. Consequently, access to some of the new emerging anti-Aβ drugs is currently restricted according to current licensing rules, depending on whether they have more than one copy of the *APOE* ε4 allele as described.

In contrast, in relation to cancer, variation in *APOE* does not appear to feature strongly in genetic risk factors for different types of cancer from various GWASs [123,159,167,183,184,185]. In relation to the study of the protein, there are also some clear-cut differences between the diseases, and in each case, there are arguably considerable gaps and missed opportunities for research that now may warrant more attention.

In AD, there is compelling *APOE* evidence and strong indications of important interactions between ApoE and Aβ that vary according to ApoE isoform and perhaps in relation to various interactions with different ApoE receptors. However, despite this level of incriminating evidence, the pipeline of direct ApoE-linked interventions has been surprisingly low over the years and likely overshadowed by the significant effort to develop anti-Aβ therapies, where ApoE might have been deemed to have a secondary or indirect modifying effect. There is now a perverse irony in how the *APOE* genotype is a key determinant in some parts of the world, e.g., EU and UK, of patient access to the new anti-Aβ drugs, and as such, there is now a need to revisit research into mechanisms by which ApoE isoforms or co-factors may be giving rise to different levels of unwanted side effects to these new drugs.

Conversely, in cancer, there is an absence of strong genetic evidence of risk for *APOE* in relation to the different cancers. However, *APOE* expression and ApoE levels appear relevant to numerous cancers. In various in vitro and in vivo models, it has been demonstrated that the presence of ApoE is important where its levels are often correlated with outcomes, representing a biomarker for diagnosis and/or a measure of progression. Furthermore, as cholesterol has also been closely linked with cancer risk and outcomes and ApoE isoforms have significant effects on cholesterol markers, interrogation into the ApoE isoforms binding interactions with their receptors in cancer is required. Similarly, that ApoE, in concert with TREM2, might modulate the immunomodulatory environment, in an isoform dependent manner also warrants further attention.

In a wider overview, whilst cancer and AD share several common risk factors, including age, obesity and type 2 diabetes, hormone signaling may be a further and overlooked contributing co-factor to the inverse association between the two diseases. Hormones play an important role in both diseases, being the key drivers of growth in hormone-sensitive cancers, but playing a potentially protective role in AD [190]. Some of these hormonal effects may be mediated by *APOE*. For example, *APOE* ε4 was found to be more strongly associated with AD risk and with worse pathology and memory impairment in females than males [191,192]. In females, an interaction between *APOE* ε4 and *ESR1* (the gene encoding estrogen receptor α) has been found, with *APOE* genotype affecting levels of estrogen receptors and hence influencing estrogen signaling [193,194]. In males, low levels of circulating ‘free’ testosterone were associated with AD risk [195] and with *APOE* ε4 carrier status [196]. In cancer, the association between *APOE* and hormone effects has not been extensively investigated, yet studies have shown that *APOE* ε2 may be protective against cognitive decline following chemotherapy for breast cancer in older patients [197], whilst *APOE* ε4 carriers may be at greater risk [198].

As a response to the question posed in the title of this review, we conclude that there is sufficient evidence to support the candidacy of ApoE as a complex mediator in the pathogenesis of both AD and some forms of cancer, which may contribute to some of the inverse associations reported between the two diseases. In AD, the role of ApoE is perhaps more direct and related to its close interactions with Aβ, and naturally dependent on genetic variation in individuals. In cancer, the role of ApoE may be more subtle, complex and yet clinically significant, with strong indications that it influences aspects of the cellular microenvironment that are critical for the development of cancer. These could relate to cholesterol synthesis and transport mechanisms due to interactions between ApoE and its receptors or to immunoregulation as have been suggested in PCa, breast, and ovarian cancers. Whether there is a clear dichotomous role for ApoE and its isoforms in both diseases is less clear to answer at this point. This is partly due to the surprising lack of research focused on ApoE isoforms in various cancers, which would allow some more direct comparisons with what is known with AD.

In conclusion, ApoE function is clearly relevant and significant in both diseases. As such, drugs being developed to directly modulate ApoE function, or drugs for other conditions that may indirectly change ApoE levels and/or differential effects of ApoE isoforms will need very careful long-term assessment both in terms of safety and should be based on precise understanding of the role of ApoE in these diseases. This would hopefully help to avoid what might be unintended consequences from the well-intended use and development of any such treatments for these conditions in the future.

## Figures and Tables

**Figure 1 genes-16-00331-f001:**
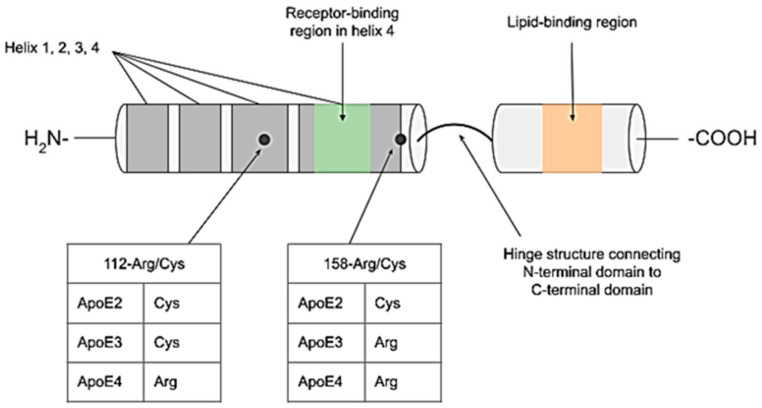
Location of the altered residues within a representation of the overall ApoE protein structure and its functional domains. The tables highlight the approximate positions of residues 112 and 158 in relation to the overall ApoE structure and indicate which amino acid residues are present at these sites in each isoform. A region in helix 4 of the amino-terminal domain marked in green denotes the presence of the receptor-binding region of the protein. The lipid-binding region is denoted in orange in the carboxyl-terminal domain.

**Figure 2 genes-16-00331-f002:**
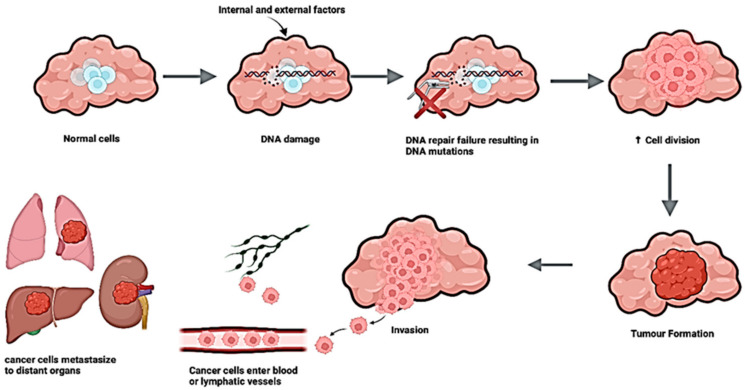
Cancer cell progression. Normal cells can undergo DNA damage due to internal or external factors, and a failure in repair of these will result in mutations, leading to uncontrolled cell growth and tumor formation. These cancer cells can invade the surrounding tissue, travel through the blood or lymphatic vessels and metastasize to distant organs. Created by Biorender.com.

**Figure 3 genes-16-00331-f003:**
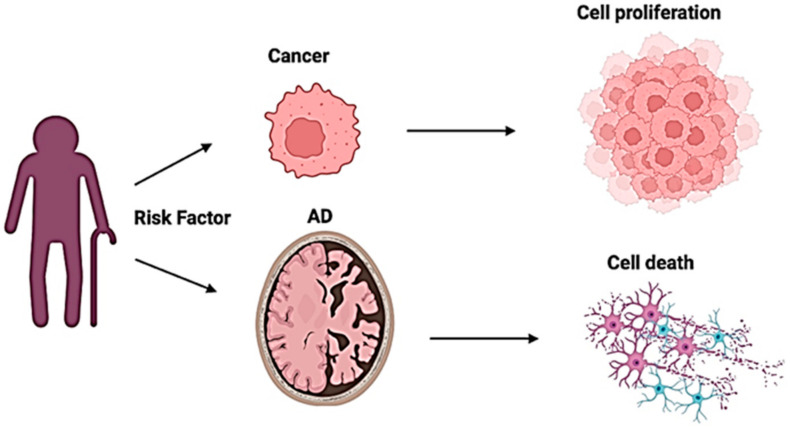
Age is a major risk factor for the development of cancer and AD; however, these diseases have opposing effects on cells. Cancer is associated with enhanced cell proliferation and survival, while AD is characterized by a lack of proliferation and neuronal cell death. Created by Biorender.com.

**Table 1 genes-16-00331-t001:** Summary of LDL receptor effects in CNS for each ApoE–Receptor complex and current understanding of isoform-specific impacts.

		Isoform-Specific Effects		
Receptor	Role of ApoE-Receptor Complex in CNS	ApoE2	ApoE3	ApoE4
Low-density lipoprotein receptor (LDLR)	Mediates uptake of lipids and cholesterol into cells; key component in cholesterol homeostasis, participates in cell signaling [9].	Extremely poor binding affinity results in decreased reuptake into cells. Associated with type III hyperlipidemia [10].	Normal binding and reuptake activity [10].	Increased binding affinity has been proposed but resulting in the “trapping” of E4 protein, leading to its decreased availability and impaired lipid uptake. Contributes to hypercholesterolemia and atherosclerosis. [11].
LDL-receptor related protein 1 (LRP1)	Aids lipid and cholesterol metabolism, functions in cell signalling, and mediates amyloid-β (Aβ) reuptake [12].	Potential protective activity from certain neurogenic diseases. Supports cell signalling pathways, positive neurotrophy, and decreased Aβ aggregation [13].	May have similar profile of functions to ApoE2 but not to a similar degree. [12].	Increased receptor binding affinity. Promotes excess accumulation of Aβ through several proposed but still unclear mechanisms [12,14].
Very-low-density lipoprotein receptor (VLDLR)	ApoE binding impacts the Reelin signalling pathway, critical in cerebellar development and adult neural plasticity. Some potential roles in relation Aβ handling in cells [15].	Mildly impairs the receptor recycling back to cell-surface following endocytosis [16].	Moderately impairs receptor recycling back to cell surface post-endocytosis [16].	Severely impairs receptor recycling back to the cell surface and reduces availability of Reelin receptors, negatively impacting neural health [16].
Apolipoprotein E receptor 2 (ApoEr2 or LRP8)	Similar role to VLDLR in the Reelin pathway, and important to cortical and hippocampal development [15].	Mildly impairs the receptor recycling back to cell-surface following endocytosis [16].	Moderately impairs receptor recycling back to cell surface post-endocytosis [16].	Severely impairs receptor recycling back to the cell surface and reduces availability of Reelin receptors, negatively impacting neural health [16].
Megalin (LRP2)	Supports endocytic uptake of lipids e.g., cholesterol, and promotes regenerative and neuroprotective functions, implicated in Aβ clearance from cells [17,18].	Research on ApoE E2-Megalin interactions is very limited.	Research on ApoE E3-Megalin interactionss is very limited.	ApoE E4-Megalin shown to hinder Aβ clearance from cells, but the mechanism is unknown [19].
Low-density lipoprotein receptor-related protein 4 (LRP4)	Astrocytic LRP4 shown to promote uptake of Aβ into astroctyes by binding ApoE [20].	ApoE E2-LRP4 interactions have not currently been extensively studied, thus data is limited.	Higher binding affinity results in “normal” Aβ uptake activity [20].	Reduced binding affinity compared to ApoE3 yields lower Aβ uptake and thus reduced Aβ-clearance via astrocytes [20].
Low-density lipoprotein receptor-related protein 1b (LRP1b)	Possible role in endocytic metabolism of ApoE-bound lipoproteins (i.e., cholesterol)—lower expression/ limited tissue distribution compared to other LDL receptors suggests a less important role or more specific role, e.g., such as cell signaling due to a large cytoplasmic tail [21].	ApoE2-LRP1b interactions have not been extensively studied, thus data is limited.	ApoE3-LRP1b interactions have not = been extensively studied, thus data is limited.	Limited protein isoform information but in Parkinson’s Disease (PD), presence of the *APOE* ε4allele and *LRP1b* rs80306347 variants was associated with increased progression to PD dementia, proposed as a result of, but not tested, impaired metabolism of amyloid precursor protein (APP) [22].
LR11/SorLA	Mediates cellular uptake of both ApoE and Aβ in an ApoE isoform-dependent manner.	No enhancement of uptake of ApoE E2 and Aβ by LR11/SorLA.	Enhanced uptake of ApoE E3 and Aβ associated with LR11/SorLA.	Enhanced uptake of ApoE E4 and Aβ associated with LR11/SorLA [23].

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
