# Peer review of "Curious Dichotomies of Apolipoprotein E Function in Alzheimer’s Disease and Cancer—One Explanatory Mechanism of Inverse Disease Associations?"

_genes, 2025, doi:10.3390/genes16030331_

Round 1

Reviewer 1 Report

Comments and Suggestions for Authors

This is a very good review paper and gives a very nice overview of the role of ApoE in cancer and Alzheimer's disease.
I have the following suggestions to improve the paper:
1. Different haplotypes of APOE regulate the differential expression of TOMM40 (PMID: 33804213), which could be a potential mechanism for its impact. TOMM40 is a crucial protein for the mitochondrial functions.
2. Could the authors also provide some inheritance values for APO genotypes? How much heritability for AD is covered by APOE? And how much for other conditions that are discussed. The authors could even build a comparative table showing the differences in the proportion of heritability. APOE is considered one of the strongest heritable factors for AD. However, it would be very informative for the readers to present some numeric values.
3. Maybe the authors can also provide a table or just a paragraph about the interacting genes and proteins for APOE. This information can be too complex, but it can be very informative. 

Author Response

We thank the reviewer for taking the time to read our review and for their constructive suggestions.

I have the following suggestions to improve the paper:

  1. Different haplotypes of APOE regulate the differential expression of TOMM40 (PMID: 33804213), which could be a potential mechanism for its impact. TOMM40 is a crucial protein for the mitochondrial functions. We thank the reviewer for highlighting this publication and have included this report on line 237 as follows: In 2021, the APOE E4 haplotype, was shown to have a significant influence on the abundance of a protein critical for maintaining mitochondrial function, the mitochondrial import receptor subunitTOMM40 (99)
  2. Could the authors also provide some inheritance values for APO genotypes? How much heritability for AD is covered by APOE? And how much for other conditions that are discussed. The authors could even build a comparative table showing the differences in the proportion of heritability. APOE is considered one of the strongest heritable factors for AD. However, it would be very informative for the readers to present some numeric values.

Thank you for this suggestion. It seems apparent that heritability values for APOE variants in relation to AD are extremely variable as it is influenced by many factors. Furthermore, information on this relating to cancer is very limited as we tried to highlight in our conclusions, despite the interesting data around the measurement of ApoE protein levels in blood in some cancers. We therefore feel, given the complexity of this subject, not only in AD but also the significant gaps in knowledge on this in relation to cancer, it is beyond the scope of our review since an additional table would be insufficient, without extensive supportive discussion, to assimilate such variable data. However, it is an excellent point, and we have added the following text and citations to recognise this comment on line 228:

Notably, whilst heritability estimates for APOE variants for AD have been determined in many different studies, they are very variable due to the direct and indirect influence of many different factors {96, 97}

  1. Maybe the authors can also provide a table or just a paragraph about the interacting genes and proteins for APOE. This information can be too complex, but it can be very informative.

Again, this is another excellent suggestion, but we again feel it is beyond the scope of this review. The interactomes, which are indeed very important, will likely be cell-type specific and as we discuss the brain and a range of peripheral cancers (which will have different specific cell types) in the review, this will be a large and complex range of information to relay, as indicated by the reviewer. We explored this to some extent by discussing the variety of ApoE receptors that exist and begin to demonstrate some of the complexity but this interesting question posed could lend itself to another informative review, but which would need to be quite extensive to do it proper justice.

Reviewer 2 Report

Comments and Suggestions for Authors

In the article, the authors discuss the role of apolipoprotein in Alzheimer’s disease and cancer pathophysiology. They explore how understanding the mechanism of action of ApoE in these contrasting diseases could enable the development of targeted therapeutics, ultimately leading to beneficial outcomes for both conditions.

Scientific suggestions

  1. Are there any FDA-approved drugs targeting ApoE for cancer and Alzheimer's disease treatment? If so, please provide a summary.
  2. Adding a summary table of approved or ongoing clinical trials would enhance readability and help readers understand the information more easily.

Minor comments/typo error

  1. Check and correct the citation format in line 86.
  2. Enhance the resolution of Figures 1 and 2.
  3. Expand the Figure 3 legend by adding one more line to describe the opposing effects on the cells.
  4. Correct "ɣ-secretase" in line 213.
  5. Modify the reference style to match the journal's formatting.

Author Response

We thank the reviewer for taking the time to read our review and for their constructive suggestions.

In the article, the authors discuss the role of apolipoprotein in Alzheimer’s disease and cancer pathophysiology. They explore how understanding the mechanism of action of ApoE in these contrasting diseases could enable the development of targeted therapeutics, ultimately leading to beneficial outcomes for both conditions.

Scientific suggestions

  1. Are there any FDA-approved drugs targeting ApoE for cancer and Alzheimer's disease treatment? If so, please provide a summary.
  2. Adding a summary table of approved or ongoing clinical trials would enhance readability and help readers understand the information more easily. Thank you for these questions.

There are no licensed treatments targeting  ApoE in either AD or Cancer, which at the very least might be a surprise for AD given the volume of evidence.  and only one in Phase II studies (please see lines 262-269). Furthermore, our discussion of new emerging drugs for AD (lines 266-289) touches on what might be indirect effects of APOE genotype, and likely ApoE isoform functional differences in mechanisms associated with the mechanisms of action of these drugs. Beyond these there is surprisingly little else to be able to summarise for interested readers.

Minor comments/typo error

  1. Check and correct the citation format in line 86.
  2. Enhance the resolution of Figures 1 and 2.
  3. Expand the Figure 3 legend by adding one more line to describe the opposing effects on the cells. Thank you-amended.
  4. Correct "ɣ-secretase" in line 213.

Modify the reference style to match the journal's formatting.

All amended as suggested-thank you for noticing these errors.